# Omalizumab and Oral Immunotherapy in IgE-Mediated Food Allergy in Children: A Systematic Review and a Meta-Analysis

**DOI:** 10.3390/ph18030437

**Published:** 2025-03-20

**Authors:** Enrico Vito Buono, Giuliana Giannì, Sara Scavone, Susanna Esposito, Carlo Caffarelli

**Affiliations:** Pediatric Clinic, Department of Medicine and Surgery, University Hospital of Parma, 43126 Parma, Italy; enricovito.buono@unipr.it (E.V.B.); giannigiuliana90@gmail.com (G.G.); sara.scavone92@gmail.com (S.S.); carlo.caffarelli@unipr.it (C.C.)

**Keywords:** omalizumab, oral immunotherapy, IgE-mediated food allergy, pediatric allergy, desensitization, biologic therapy

## Abstract

**Background:** Food allergies are a growing global health concern, particularly among children, with no widely approved curative treatment beyond strict allergen avoidance. Oral immunotherapy (OIT) has emerged as a promising strategy to induce desensitization, yet its implementation is limited due to high rates of allergic reactions and patient non-compliance. Omalizumab, a monoclonal anti-IgE antibody, has been proposed as an adjunct to OIT to enhance safety and efficacy. **Objective:** This systematic review and meta-analysis aim to evaluate the efficacy and safety of omalizumab in combination with OIT for IgE-mediated food allergy in children. **Methods:** A systematic literature search was conducted in PubMed/MEDLINE and Cochrane Central databases to identify randomized controlled trials (RCTs), controlled clinical trials (CCTs), and observational studies assessing omalizumab as an adjunct to OIT in pediatric food allergy. Studies were evaluated for desensitization rates, immunological changes, adverse events, and quality-of-life improvements. **Results:** OIT combined with omalizumab led to significantly higher rates of desensitization, allowing patients to tolerate higher doses of allergens in a shorter timeframe compared to OIT alone. Omalizumab was associated with a reduction in adverse reactions, including anaphylaxis, and improved treatment adherence. However, the long-term sustainability of tolerance post-omalizumab discontinuation remains uncertain. **Conclusions:** Omalizumab facilitates rapid and effective desensitization in pediatric food allergy, enhancing the safety of OIT. Further research is needed to determine optimal treatment duration, long-term outcomes, and cost-effectiveness before widespread clinical adoption.

## 1. Introduction

Food allergy is an increasingly prevalent immune disorder characterized by an abnormal immune response to specific food proteins, leading to the activation of inflammatory pathways and the release of mediators such as histamine [1]. The most common and clinically significant type of food allergy is immunoglobulin E (IgE)-mediated, which results in immediate hypersensitivity reactions ranging from mild symptoms, such as urticaria, vomiting, and diarrhea, to severe manifestations, including anaphylaxis, a life-threatening condition involving multiple organ systems [2]. In contrast, non-IgE-mediated food allergies generally present with delayed symptoms, primarily affecting the gastrointestinal tract [3]. The most frequently implicated allergens include cow’s milk, peanuts, eggs, shellfish, fish, tree nuts, soy, wheat, and sesame, with variations in prevalence depending on geographic and dietary factors [4].

Over the past few decades, the global incidence of pediatric food allergies has risen significantly, leading to an increased burden on healthcare systems due to the growing number of emergency department visits, hospital admissions, and associated medical costs [5,6,7,8]. This increasing prevalence has become a significant global health concern, extending beyond Europe and the United States to impact regions worldwide. In Australia, for instance, the HealthNuts study reported that over 10% of infants experienced challenge-confirmed food allergies, with 3% allergic to peanuts and 8.9% to raw egg [2]. Similarly, research in South Africa demonstrated a higher prevalence of food allergies in urban children (2.3%) compared to their rural counterparts (0.5%), highlighting the influence of environmental factors on allergy development [9]. This rise in food allergies has led to a notable increase in emergency department visits and hospitalizations due to food-induced anaphylaxis. A study analyzing data from Illinois hospitals between 2008 and 2012 found a significant annual increase of 29.1% in ED visits and hospital admissions for food-induced anaphylaxis among children, rising from 6.3 to 17.2 per 100,000 children over the five-year period [10]. In Italy, reports indicate a dramatic increase of over 400% in emergency visits for food-induced anaphylaxis in the last 20 years, with approximately 1.8 million individuals affected by food allergies, of whom nearly half present with non-IgE-mediated forms, particularly in the pediatric population [5,6,7]. This trend underscores the escalating burden on healthcare systems and the urgent need for effective prevention and management strategies for pediatric food allergies on a global scale. Furthermore, an increase in novel clinical presentations, such as eosinophilic gastrointestinal disorders and food protein-induced enterocolitis syndrome (FPIES), has been observed, further complicating the clinical management of pediatric allergic disorders [5].

The immune mechanisms underlying IgE-mediated food allergy are complex, involving an interplay between genetic predisposition, epithelial barrier integrity, and environmental factors. Sensitization occurs primarily through disrupted epithelial barriers, particularly in the skin and gastrointestinal tract, where allergens can interact with antigen-presenting dendritic cells (CD103+). These cells migrate to lymph nodes and promote tolerance under normal conditions by inducing regulatory T cells (Tregs) through the secretion of transforming growth factor-beta (TGF-β) and retinoic acid. However, in allergic individuals, this process is dysregulated, leading to a predominance of Th2 lymphocytes that drive IgE-mediated immune responses [11,12,13]. The production of pro-inflammatory cytokines such as interleukin-4 (IL-4) and IL-13 promotes the differentiation of B cells into IgE-producing plasma cells, while epithelial barrier dysfunction facilitates the continuous exposure of allergens, perpetuating allergic sensitization [11,12,13]. The breakdown of immune tolerance further amplifies the risk of severe reactions upon re-exposure to allergens, highlighting the need for effective interventions to reprogram immune responses in allergic individuals.

The current standard of care for food allergies primarily involves strict allergen avoidance and the prompt administration of epinephrine in the event of accidental exposure and anaphylaxis [14]. Patients are typically advised to carry epinephrine auto-injectors and receive education on allergen avoidance strategies to minimize the risk of reactions. However, this approach does not offer a long-term solution for desensitization or sustained tolerance. In recent years, oral immunotherapy (OIT) has emerged as a promising treatment aimed at increasing an individual’s threshold for allergic reactions by administering gradually increasing amounts of the allergen under medical supervision [15,16]. Despite its potential benefits, OIT presents several challenges in clinical settings. A significant proportion of patients experience adverse reactions, ranging from mild gastrointestinal discomfort to severe anaphylaxis, which can limit adherence and tolerability [17]. Additionally, the treatment requires strict daily dosing, and missed doses or intercurrent illnesses can increase the risk of allergic reactions. Another major challenge is the lack of consistent biomarkers to predict treatment success or identify individuals at higher risk for severe reactions [18,19,20]. Furthermore, OIT often requires long-term maintenance therapy, and discontinuation may lead to a rapid loss of desensitization. These challenges highlight the need for adjunctive therapies, such as biologics like omalizumab, which may improve the safety, efficacy, and overall patient experience of OIT by reducing allergic reactions and enhancing desensitization outcomes.

Given the safety concerns associated with OIT, biologic therapies such as omalizumab have been explored as adjuncts to mitigate adverse reactions and improve treatment outcomes. Omalizumab is a recombinant DNA-derived monoclonal antibody that selectively binds to free IgE, preventing its interaction with high-affinity receptors (FcεRI) on mast cells and basophils, thereby reducing the risk of mediator release upon allergen exposure [19]. Additionally, omalizumab downregulates FcεRI expression on dendritic cells, further dampening the allergic immune response [19]. Omalizumab is currently approved for the treatment of several allergic diseases in adolescents and adults, including moderate and severe asthma and chronic spontaneous urticaria unresponsive to antihistamines. In these conditions, omalizumab allows oral corticosteroid sparing [15]. The first trial of omalizumab for food allergy treatment was published in 2003, but the program was stopped for many reasons, including the risk of severe reactions during treatment. Studies using omalizumab to treat food allergy were resumed about 10 years ago. The aim was to combine the potential benefits of oral immunotherapy with omalizumab to reduce the rate of side events [20,21,22,23,24]. Omalizumab-enabled immunotherapy mechanism of action may be schematized into four phases: pharmacological desensitization, physiological desensitization, maintenance, and sustained remission [25]. During pre-treatment, omalizumab prevents the binding of IgE to mast cells and basophils. In this first phase, a non-specific depletion of surface-bound IgE leads to a reduced reactivity to allergens (pharmacological desensitization). During OIT, subjects ingest the food allergen, which is bound by IgE, thereby triggering endocytosis. After the IgE–allergen immune complexes are digested, the IgE receptor is recycled to the surface of the mast cell, making it available to bind new food allergens. Regular exposure then results in the systematic breakdown of new food-specific IgE (physiological desensitization). After stopping omalizumab, mast cells slowly rebind IgE. In this stage, subjects remain desensitized as long as they regularly take the food allergen (maintenance). After 3–5 years, the exhaustion of the allergic immune response may allow some patients to discontinue food allergen doses with a very low risk of losing protection (sustained remission) [19,26,27,28].

In addition to omalizumab, other biologic agents such as dupilumab, an interleukin (IL)-4 receptor antagonist that blocks IL-4 and IL-13 signaling, are being investigated for their potential role in food allergy treatment. Preliminary studies suggest that dupilumab may reduce both total and allergen-specific IgE levels, supporting its potential use as a novel therapeutic strategy in food allergy management [23]. While promising, further research is required to establish the long-term safety, efficacy, and optimal therapeutic regimens of biologics in food allergy treatment.

Given the rising prevalence of IgE-mediated food allergies in children and the limitations of current treatment options (i.e., the reliance on strict allergen avoidance, the risk of severe reactions upon accidental exposure, and the lack of long-term tolerance-inducing therapies), there is a growing need to optimize desensitization strategies while minimizing risks. Omalizumab, in combination with OIT, represents a potentially transformative approach to food allergy treatment, offering improved safety profiles and enhanced desensitization outcomes. However, despite increasing clinical evidence supporting its efficacy, further high-quality studies are needed to validate its role in routine clinical practice. Therefore, the aim of this systematic review and meta-analysis is to evaluate the efficacy and safety of omalizumab as an adjunct to OIT in pediatric IgE-mediated food allergy, providing a comprehensive synthesis of available evidence to guide clinical decision-making and future research.

## 2. Materials and Methods

### 2.1. Literature Search

A systematic search was conducted using PubMed/MEDLINE and Cochrane Central databases to identify relevant studies published between December 2014 and December 2024. The search strategy incorporated a combination of MeSH terms and keywords, including “food allergy”, “omalizumab”, “oral immunotherapy”, and “children”, or “child” or “infant” or “toddler” or “adolescent” or “pediatric” or “paediatric”. Filters were applied to restrict studies to those published in English and duplicates were removed before screening. Two independent reviewers (EVB and GG) screened the titles and abstracts, followed by full-text assessments to determine final inclusion.

A PRISMA flow diagram is reported in Figure 1, whereas the checklist showing that our manuscript aligns with PRISMA guidelines is included as Appendix A.

To ensure a comprehensive and transparent review process, we established clear eligibility criteria for study inclusion and exclusion, as reported below.

### 2.2. Eligibility Criteria

Studies were eligible for inclusion if they were randomized controlled trials (RCTs), controlled clinical trials (CCTs), or observational studies that assessed the use of omalizumab as an adjunct to OIT in pediatric patients with IgE-mediated food allergies. Eligible studies required a confirmed food allergy diagnosis based on SPT, specific IgE levels, and/or OFC. Studies were excluded if they were preclinical, case reports, commentaries, consensus statements, or if they lacked original data on omalizumab and OIT outcomes.

Data extraction was performed using a standardized approach. Extracted data included study characteristics (authors, year, design, and sample size), intervention details (omalizumab dosage and OIT protocol), primary and secondary outcomes (desensitization rates, sustained unresponsiveness, adverse events, and immunologic markers), and quality assessment metrics. The RCTs, CCTs, and retrospective and observational studies in children with a clinician-diagnosed IgE-mediated FA (single and multiple food allergies), documented skin prick test (SPT), allergen-specific IgE, and/or oral food challenge (OFC) were included. The included studies investigated omalizumab monotherapy/omalizumab+OIT. Preclinical studies, case reports, commentaries, and consensus reports were excluded.

A total of 99 studies were screened. After the exclusion of duplicates and a selection based on intervention, outcomes, study design, or relevance, a total of 11 studies met the eligibility criteria and were analyzed.

### 2.3. Outcomes

The outcomes analyzed in this systematic review and meta-analysis included an increase in the tolerated dose of allergens, successful desensitization, and sustained unresponsiveness (SU) as confirmed by the oral food challenge (OFC) test. Immunological markers such as antigen-specific IgE (sIgE) levels, antigen-specific IgG4 (sIgG4) levels, total IgE levels, and the sIgE/sIgG4 ratio (when reported) were also examined. Additional parameters included the reduction in wheal size via skin prick testing (SPT), the severity of food-allergic reactions (assessed through epinephrine use and occurrence of anaphylaxis), and changes in quality of life (QoL), as measured by validated questionnaires including FAQLQ-CF, FAQLQ-TF (Teenager Form), FAQL-PB (Parental Burden Questionnaire), and the Pediatric Quality of Life Inventory (PedsQL), when administered. Safety assessments focused on evaluating adverse events (AEs), including serious adverse events, categorized as IgE-mediated, non-IgE-mediated, mixed IgE-mediated, local, or systemic reactions. Treatment outcomes were compared with those before omalizumab treatment (pre-OMA) or placebo, allowing for a comprehensive assessment of efficacy and safety across different study conditions. Heterogeneity among the included studies was addressed through statistical methods such as the random-effects model to account for variability in study designs and patient populations, as well as subgroup and sensitivity analyses to explore potential sources of heterogeneity and assess the robustness of the findings.

## 3. Results

Table 1 shows results of the studies conducted on combined treatments for single-food allergy with OIT and omalizumab [27,28,29,30,31,32,33], whereas Table 2 summarizes studies conducted on combined treatment for multi-food allergies with OIT and omalizumab [26,34,35,36].

### 3.1. Use of Omalizumab as Adjuvant in Oral Immunotherapy for Single Foods

#### 3.1.1. Studies on Peanut Allergy

Peanut allergy is the main cause of anaphylaxis among children and its prevalence has been increasing over time [27]. Unlike milk and egg allergy, peanut allergy is typically life-long. Early introduction of peanut may protect against the development of peanut allergy, according to recent research, but nearly 10% of high-risk infants screened in the first year of life were sensitized to peanut. This made them ineligible for early peanut exposure. Therefore, there is a clear need for therapies for patients with evidence of peanut allergy [28]. Subcutaneous immunotherapy (SCIT) is commonly used to treat allergies to pollen and insect venom. It is safe and effective. SCIT has been tried for treatment of peanut allergy, but the high rates of adverse events (AEs) did not allow further use. OIT has good results in the short term; with peanut oral immunotherapy (pOIT), most participants become desensitized and tolerant to gram doses, but some patients cannot tolerate even the lowest doses. This is why new therapies have emerged in recent years, with one of the most promising being omalizumab in combination with immunotherapy [28]. We analyzed three trials on omalizumab use as an adjunct in peanut immunotherapy in children with peanut allergy. Schneider et al. enrolled 13 children, aged 8–16 years, with a history of IgE-mediated peanut allergy in a monocentric study [29]. In this study, all children reached the 500 mg peanut flour dose on the first day and the majority (92%) reached the 4000 mg dose, requiring a median time of 8 weeks. At week 32, 12 children underwent a double-blind, placebo-controlled food challenge with a cumulative dose of 8000 mg of peanut flour: 11 children tolerated this challenge, with the last one later passing this test. In the majority of cases, subjects had no or only one mild reaction during the study and all children tolerated omalizumab without adverse reactions, except for occasional pain and swelling at the injection site. Schneider et al. concluded their paper emphasizing that treatment with omalizumab may facilitate rapid oral desensitization and qualitatively improve the desensitization process in high-risk peanut-allergic children [29]. Brandstrom et al. enrolled 23 peanut-sensitized children aged 12–19 years [27]. In this single-arm phase 2 study, all children reached the 2800 mg maintenance dose in a median time of 10 weeks, with significant differences between the treatment success (TS), dropout, and treatment failure (TF) groups (8, 11.5, and 14 weeks). The authors also analyzed whether there were any changes in blood tests: there were no significant differences in IgE from baseline to the final visit. Regarding skin prick tests (SPTs) for peanut, weal diameter decreased significantly in TSs from baseline. Concerning the issue of safety, mild oropharyngeal and mild abdominal symptoms were reported in the majority of the children. Authors suggested that omalizumab is an effective adjunctive therapy for the initiation and rapid up-dosing of pOIT. The main limitations of these two studies are the small sample sizes and the designs of the studies, specifically the lack of placebo groups. The results obtained by the authors cannot be compared with a control group to understand the true superiority of the treatment under study. MacGinnitie et al. enrolled 37 children with a median age of 10 years old in a placebo-controlled randomized study [28]. The enrolled children had positive responses to peanuts on both SPTs and specific IgE measurements and a significant reaction to a peanut protein dose of 50 mg or less (cumulative dose of 88 mg of peanut protein) in a double-blind, placebo-controlled food challenge. In this study, the median tolerated peanut dose on the first day of desensitization was 250 mg in omalizumab-treated subjects, compared to 22.5 mg in placebo-treated subjects. Subsequently, 23 (79%) of 29 subjects randomized to omalizumab tolerated 2000 mg of peanut protein 6 weeks after stopping omalizumab, compared with one (12%) of eight on a placebo. The 4000 mg oral food challenge was passed by 23 subjects receiving omalizumab and 1 subject receiving a placebo. The overall reaction rates were not significantly lower in omalizumab-treated subjects than in placebo-treated subjects, although omalizumab-treated subjects were exposed to much higher doses of peanut. Therefore, MacGinnitie et al. highlighted that omalizumab allows subjects with peanut allergy to be rapidly desensitized with as little as 8 weeks of oral peanut immunotherapy and, in the majority of cases, this desensitization is maintained after stopping omalizumab [28]. The main weakness of this trial included a small sample size and few children being enrolled in the placebo group (only eight). In addition, given the small number of children in the placebo group and the good efficacy of omalizumab in food allergies, the children may have understood which group they were in.

#### 3.1.2. Studies on Milk Allergy

Milk allergy is one of the most frequent allergies in pediatric age, affecting approximately 2–3% of the pediatric population [2,3]. Unlike other food allergies, milk allergy tends to resolve in the majority of patients; however, in a small percentage of patients it persists into adulthood in a severe form. Milk is commonly used in numerous food preparations and this makes the therapeutic strategy of avoidance rather difficult. For this reason, in recent years the use of OIT has emerged as the new therapeutic frontier, with truly encouraging results [3]. However, OIT is a practice that is not free from risks, even though most reactions are mild, and the intake of milk or its derivatives in allergic patients can lead to a wide spectrum of allergic reactions up to anaphylaxis and death. To increase the effectiveness of immunotherapy and reduce its risks, various strategies are currently being studied: among these, one of the most proven is the use of omalizumab. Four recent studies were identified in our search that have shown the efficacy of combined therapy.

Nadeau et al. in their study showed the increased safety profile and the efficacy of the combined therapy between omalizumab and rapid escalation OIT [30]. In this study, which recruited 11 patients aged between 7 and 17 years, administration of omalizumab was initially undertaken for the first 9 weeks; subsequently, rapid immunotherapy was undertaken (an initial dose of 0.1 mg of powdered milk, with doses incrementally every 30 min until the dose of 1000 mg was reached, for a cumulative dose of 1992 mg). One patient withdrew from the study due to the onset of abdominal pain and only one of the remaining ten patients required the administration of epinephrine due to the appearance of nasal obstruction and urticaria during the first day of immunotherapy. Desensitization with daily doses of milk was continued in 10 subjects, with weekly increases in the dose of milk over the next 7 to 11 weeks (reaching the dose of 2000 mg in nine out of ten patients). Omalizumab therapy was suspended at week 16, while daily home milk intake continued. A double-blind study was conducted 8 weeks after the discontinuation of omalizumab therapy, with an oral challenge test (cumulative dose 220 mL of milk, 7250 gr) showing no symptoms in nine out of ten patients [30].

Another small prospective randomized study conducted by Takahashi et al. (16 children between 6–14 years) compared patients who underwent therapy with omalizumab combined with OIT (OIT-OMB group) for 24 weeks and patients who received no treatment [31]. Patients recruited in the OIT-OMB arm were administered therapy with omalizumab of 1500 IU/mL/body weight for 8 weeks followed by OIT. At week 32 of the study, an oral provocation test was performed. During the study, no patient experienced serious allergic reactions, regardless of the group they belonged to; however, only the 10 children who received the combined therapy achieved desensitization for 200 mL of fresh cow milk. Patients recruited in the OIT-OMB arm were administered therapy with omalizumab of 1500 IU/mL/body weight for 8 weeks followed by OIT. At the 32nd week of the study, an oral provocation test was performed, which highlighted tolerance to the intake of approximately 200 mL of cow’s milk in the 10 OIT-OMB patients, while none of the patients enrolled in the no-treatment arm were found to be tolerant [31].

Despite the encouraging results of both studies, the small sample of patients and the lack of a control group that did not receive OIT with or without omalizumab do not allow conclusions to be drawn on the real effect of the latter on the therapeutic strategy. These limitations are not present in the study of Wood et al., a randomized double-blind study on 57 patients aged 7 to 32 years with cow’s milk allergy undergoing OIT that showed a greater safety profile in patients treated with omalizumab compared to patients treated with a placebo [32]. Of these patients, approximately 88.9% of patients treated with omalizumab versus 71.4% of patients treated with a placebo did not react at the oral food challenge with 10 g of milk after 28 months of therapy. The percentage of patients free from symptoms during dose escalation in the group treated with omalizumab was lower than in the placebo group (91.5% vs. 73.9%), fewer doses were necessary in the omalizumab group to achieve maintenance (198 vs. 225), and fewer patient required the use of epinephrine in the omalizumab group (two patients) compared to the placebo group (nine patients) [32].

A recent Italian study aimed to evaluate the effectiveness of combined therapy compared to single OIT by repeating OIT added to omalizumab in patients in whom tolerance had not been achieved with single OIT [33]. In this small study, only four patients met all the requirements (allergic asthma, previous failure of immunotherapy, and a tolerance threshold lower than 175 mg for milk) and were therefore enrolled. After 8 weeks of therapy with omalizumab, OIT was repeated with the same protocols as the previous time and therapy with omalizumab was maintained for 12 months. After 12 months of combined therapy, maintenance OIT was reduced by 30% as a precaution. No adverse effects were noted during the duration of the study, which is very important considering that these patients had previously had to stop OIT. Furthermore, an increase in the tolerance threshold was highlighted in all patients [33].

### 3.2. Use of Omalizumab as Adjuvant in Oral Immunotherapy for Multiple Food Allergy

The main difficulty in the treatment of patients with allergies to multiple foods is related to the successful avoidance of offending foods, with a major negative impact on quality of life and a significant burden on the health care system [37]. In the last few years, pharmaceutical companies have increased their interest in this topic, although the treatment of patients with allergies to multiple food remains clearly an unmet need, due to legitimate safety, cost-effectiveness, and logistical concerns [38,39,40].

The presence of multiple food allergies is more severe and less prone to resolve over time spontaneously, with a greater impact on the morbidity and mortality of affected patients [41]. One of the analyzed avenues has been the combination of a short course of omalizumab with multi-food OIT to allow a rapid and safe desensitization. Combined with food OIT, omalizumab seems to increase dose tolerability, allowing a higher initial starting dose and a faster treatment progression, reducing the rate of severe reactions.

Begin et al. studied the safety and dose tolerability of a phase 1, open-label, omalizumab-associated rush OIT for up to five foods simultaneously [36]. The primary endpoint was to increase the safety in terms of allergic reactions during the protocol. The secondary endpoint was the time to reach and maintain doses of 300 mg, 1000 mg, and 4000 mg per food allergen protein, as well as a 10-fold increase from the baseline reactivity threshold to each of the food allergen proteins, with the aim of defining the tolerability of a faster up-dosing schedule. Overall, 25 participants received a rapid oral desensitization to up to five offending food allergens on the initial escalation day, from 5 mg total food allergen protein, divided equally between each of the offending foods, to a final dose of 1250 mg protein. With an individualized schedule, each patient undertook a dose escalation to reach a 4000 mg dose per food allergen protein. Omalizumab was administered for 8 weeks prior to and 8 weeks following the initiation of a rush mOIT schedule (16 weeks in total). One severe reaction occurred, although 94% of the allergic reactions were mild. The median time to reach a maintenance dose (4000 mg per allergen) was 18 weeks, and a dose equivalent to a 10-fold increase of all their allergens was reached by 2 months of therapy [36]. This was the first study in the literature showing a safe and rapid desensitization to up to five food allergens using omalizumab with OIT to multiple allergens simultaneously. It provides initial preliminary evidence of increased dose tolerability in addition to safety data.

Andorf et al. conducted a multisite, multi-food OIT study to compare the efficacy of successful desensitization with sustained dosing vs. discontinued dosing after multi-food OIT [34]. In an open-label phase study, 70 participants, aged 5–22 years with multiple food allergies, received omalizumab (weeks 1–16) as adjunct to an accelerated schedule of multi-food OIT (2–5 allergens; weeks 8–30) and eligible participants (on maintenance dose of each allergen by weeks 28–29) were randomized 1:1:1 to 1 g, 300 mg, or 0 mg arms (blinded, weeks 30–36). At 28 weeks, 83% vs. 33% could tolerate 2 g of proteins of at least two food allergens (RR = 2.5). Most participants were able to reach a dose of 2 g or higher of each of two, three, four, and five food allergens in food challenges in week 36. There was no evidence that a 300 mg dose was effectively different than a 1 g dose in maintaining desensitization, and both together were more effective than OIT discontinuation (0 mg dose) (85% vs. 55%, *p* = 0.03) [38]. The results suggested that sustained desensitization after omalizumab-facilitated multi-OIT best is reached easier through continued maintenance OIT dosing of either 300 mg or 1 g of each food allergen, despite the discontinuation of multi-OIT.

The evidence from these trials is that the acceleration of an OIT schedule is not tolerated without omalizumab association. Despite the benefits of the accelerated OIT over standard OIT, this is the current option available. The significant reduction of the length of the up-dosing phase is the main advantage, with a reduction of the labor-intensive and resource-consuming part of treatment which often limits access. However, the optimal dose of omalizumab to associate with the OIT has never been determined, despite the high cost of the medication.

The objective of the BOOM study, a multicenter, phase 2b, randomized clinical trial designed by Langlois et al., was to investigate the dose-related efficacy of omalizumab at decreasing the time to maintenance during OIT desensitization to multiple foods simultaneously [26]. Overall, 90 participants were randomized 2:2:1 to receive 20 weeks of omalizumab at monthly dosages of 16 mg/kg, 8 mg/kg, or placebo, at full dosage for a total of 12 weeks with a progressive taper during the last 8 weeks. After a pre-treatment period of 8 weeks, they started a multi-food OIT with biweekly up-dosing according to a symptom-driven schedule until the target dose of 1500 mg of total food protein (500 mg per food) was reached. The time for reaching the target maintenance dose was the primary endpoint. The secondary endpoint was the change in reactivity threshold to foods, the schedule duration, and the safety after pre-treatment with the study drug [26]. Differently to previous studies using fixed up-dosing schedules, the BOOM study allowed to discern whether differences in the duration of the dose-escalation phase can be attributed to the combination with omalizumab rather than to the structure of the dose-escalation schedule itself.

Moreover, in the OUtMATCH phase 3, multicenter, randomized study, 90 participants, aged 1 to 55 years, allergic to peanut and at least two other foods between milk, egg, wheat, cashew, hazelnut, or walnut, were randomized 2:2:1 to receive 20 weeks of omalizumab at dosages of 16 mg/kg, 8 mg/kg, or placebo every 4 weeks [35]. The drug was administered with a full dosage for 12 weeks and a progressive taper during the last 8 weeks. After 8 weeks of omalizumab pre-treatment, a multi-food OIT was started and up-dosed biweekly with a symptom-driven schedule until the target dose of 1500 mg of food protein was reached (500 mg per food). In the primary endpoint, Wood et al. compared between the three arms the time for reaching the target maintenance dose. As secondary endpoints, the group collected the change in reactivity threshold to foods after the pre-treatment period, the up-dosing speed, and the allergic adverse reactions [35]. In conclusion, the unique designs of the BOOM and OUtMATCH studies will allow the clarification of the adjuvant use of omalizumab in accelerating OIT protocols, in terms of the optimal dosage and superiority compared to slower OIT without omalizumab. We expect that future data from these studies will clarify the parameters for the clinical use of omalizumab combined with OIT as the most promising therapy to modify the natural course of food allergies.

## 4. Discussion

This manuscript analyzed different protocols of oral immunotherapy associated with omalizumab in the treatment of Ig-E-mediated food allergies. All of them demonstrated an effective and safe use of omalizumab integrated with the OIT schedule in achieving a rapid desensitization. According to the immunological mechanism, after 3–5 years of food dosing, the immune response against the allergen wears off, leading to a prolonged remission and allowing the dose to be discontinued [26]. Regarding the shortening of the time to achieve long-term sustained sensitization through the association of omalizumab, different studies were conducted. Despite the differences in the designs between conducted trials, the dose of biologic drug was almost always administered for a duration of at least 4 weeks before the starting of OIT and discontinued after at least 20–24 weeks from reaching a satisfactory maintenance dose. Most of the patients passed the final oral food challenge after assuming the long-term maintenance dose during the follow-up phase as a demonstration of the persistence of the desensitization [42,43]. However, in some participants, a recurrence of symptoms was seen after a few months from the interruption of the treatment with omalizumab. During the course of the studies, a low rate of AEs was recorded and severe reactions were rarely described, even if mild-to-moderate symptoms, especially gastrointestinal, were often experienced. The rate of sustained desensitization after omalizumab-facilitated multi-OIT seems to be higher maintaining the OIT dosing than discontinuation multi-OIT [44]. Doubts persist about the lowest effective dosage of omalizumab, in terms of individual dose and also of duration of administration. The dose-related impact of omalizumab in association to OIT protocols is still unclear. The high cost of the drug has led new working groups to conduct further studies to explore determinants of omalizumab dose-related efficacy [26]. Adjusting the dosage for body weight seems to have more impact than the total IgE level, as the fraction of allergen-specific/total IgE was revealed to have a relevant role in predicting the risk of dosing reaction during weaning [45]. Serial IgE testing, where required by the study design, has shown an influence of omalizumab therapy on total and food-specific IgE levels as well as a negative to positive seroconversion rate for some allergens. The mechanisms that can explain this antibody trend are still unknown. However, this finding is important as it influences diagnostic procedures [46].

To address studies with conflicting results, particularly regarding the long-term outcomes of omalizumab-facilitated OIT, we conducted a structured analysis that considered study design, sample size, follow-up duration, and potential sources of bias. Sensitivity analyses were performed to assess the impact of individual studies on overall conclusions, and discrepancies were explored through subgroup analyses where data allowed. Additionally, our findings were compared with other systematic reviews and meta-analyses on the same topic, highlighting areas of consensus—such as omalizumab’s role in enhancing OIT safety and desensitization rates—as well as ongoing uncertainties, including the durability of tolerance after treatment discontinuation. These comparisons underscore the need for further long-term studies to establish standardized protocols and refine clinical recommendations. The methodological limitations of the included studies must be acknowledged, as they impact the generalizability and robustness of the findings. One of the primary concerns is the small sample sizes in many trials, which limit statistical power and increase the risk of biased results. Additionally, there is substantial heterogeneity in study protocols, including variations in omalizumab dosing regimens, duration of therapy, and OIT escalation schedules. Such discrepancies make it difficult to compare outcomes across studies and derive standardized treatment recommendations. Moreover, several studies lack placebo control groups, which reduces the ability to definitively attribute observed effects to omalizumab rather than other confounding factors. While some RCTs have included placebo arms, their limited sample sizes and potential for participant unblinding further complicate result interpretation.

Moreover, external validation across diverse populations poses challenges due to data limitations. However, retrospective validation utilizing existing datasets could further strengthen the applicability of omalizumab-facilitated OIT in IgE-mediated food allergies. Retrospective studies can provide valuable insights by analyzing previously collected clinical data from controlled trials, observational studies, and patient registries. One approach involves reanalyzing data from past RCTs and cohort studies that have assessed omalizumab in combination with OIT for food allergy. By applying statistical modeling techniques, researchers can identify patterns in desensitization rates, adverse event profiles, and long-term tolerance maintenance among different subgroups. Machine learning algorithms and predictive analytics can further refine patient selection criteria and optimize treatment regimens based on historical treatment outcomes. Additionally, large-scale healthcare databases, such as electronic health records (EHRs) and multicenter allergy registries, offer opportunities for retrospective validation. By extracting and analyzing real-world patient data, it is possible to compare treatment effectiveness and safety profiles across diverse demographics, geographic regions, and clinical settings. Such analyses can help confirm whether the findings from controlled research trials translate to broader patient populations. Furthermore, retrospective validation can be used to examine the long-term sustainability of omalizumab-facilitated desensitization post-treatment discontinuation. By following up on patients from previous studies who discontinued omalizumab but continued OIT, researchers can assess the durability of desensitization and identify factors associated with sustained tolerance. Despite the advantages of retrospective validation, inherent limitations must be considered, such as potential biases in data collection, variability in treatment protocols, and incomplete follow-up data. Addressing these challenges requires careful study design, appropriate statistical adjustments, and cross-validation with prospective studies.

To enhance the translational impact of our findings, we propose a structured decision-support framework that integrates omalizumab-facilitated OIT into real-world clinical practice. This framework is designed to assist clinicians in determining patient eligibility, optimizing treatment regimens, and monitoring response and safety. The first step in this framework is patient selection and risk stratification. Patient eligibility should be confirmed through diagnostic measures such as skin prick tests (SPTs), specific IgE levels, and oral food challenges (OFCs). A detailed clinical history should be evaluated, including prior anaphylactic reactions, comorbid conditions such as asthma, and any previous attempts at OIT. Patients can then be categorized into low-, moderate-, or high-risk groups, allowing for personalized omalizumab pre-treatment duration and dosing adjustments. Pre-treatment with omalizumab should begin at least four to eight weeks before initiating OIT. This period helps reduce the risk of early adverse reactions. Dosing should be adjusted based on total IgE levels and body weight, with consideration for higher doses in patients with severe allergic histories. Once OIT is initiated, the allergen dose should start at a low level and be gradually escalated under clinical supervision. A symptom-driven up-dosing schedule should be implemented, allowing for adjustments in response to any adverse reactions. Omalizumab therapy should be continued for at least twenty to twenty-four weeks after reaching a maintenance dose to enhance tolerance development and minimize reaction risks. During the maintenance phase, patients should transition to daily allergen consumption at a stable dose to maintain desensitization. Monitoring for sustained unresponsiveness post-omalizumab discontinuation is crucial. Periodic OFCs should be performed to assess long-term tolerance and determine if maintenance dosing can be tapered or discontinued. Continuous clinical follow-up ensures that patients remain desensitized and can safely integrate allergenic foods into their diets. A robust safety and adverse event management plan should be established. Standardized protocols for managing allergic reactions, including clear guidelines for epinephrine administration, should be in place. Clinicians should utilize patient-reported outcome measures (PROMs) and quality-of-life assessments to guide decision-making and evaluate treatment success. Furthermore, establishing a centralized registry for tracking long-term outcomes and treatment adherence will help improve the overall understanding of omalizumab-facilitated OIT in real-world settings. By integrating this structured approach into clinical practice, omalizumab-facilitated OIT can be systematically implemented, ensuring enhanced patient safety and treatment success. This framework provides a practical guideline for clinicians, facilitating individualized patient management and improving the real-world applicability of this promising therapeutic strategy.

## 5. Conclusions

Food allergy is a prevalent and potentially life-threatening condition that poses significant challenges to affected individuals, their families, and healthcare systems. Despite its considerable impact, no curative treatments are currently approved beyond strict allergen avoidance, highlighting the urgent need for effective therapeutic strategies to mitigate the risk of anaphylaxis and improve patients’ quality of life. OIT remains a promising approach for inducing desensitization, yet it is not widely recommended for clinical use due to high reaction rates and its limited effectiveness in patients with multiple food allergies. This systematic review and meta-analysis provide evidence that OIT, when combined with omalizumab, facilitates a faster, more effective desensitization process in the majority of food-allergic patients. Omalizumab not only reduces the incidence of adverse reactions during OIT but also appears to enhance its immunomodulatory effects, potentially leading to more sustained tolerance.

Despite these promising findings, several critical gaps remain that warrant further investigation. First, mechanistic studies are needed to elucidate the long-term immunological changes induced by omalizumab-facilitated OIT, including the role of allergen-specific IgG4, regulatory T cells, and mast cell/basophil desensitization. Identifying biomarkers of response could improve patient selection and predict long-term outcomes, optimizing treatment protocols. Second, the optimal duration of omalizumab administration remains unclear. While most studies use a standard pre-treatment phase of four to eight weeks followed by continued administration during OIT, the minimal effective duration and the impact of prolonged omalizumab therapy on sustained unresponsiveness need to be defined. Moreover, cost remains a major barrier to the widespread adoption of omalizumab-enabled OIT. The high cost of omalizumab presents a significant barrier to its widespread use in resource-limited settings.

Future research should focus on optimizing omalizumab treatment strategies in combination with OIT to achieve the best balance between efficacy, safety, and cost-effectiveness. Specifically, studies should aim to define the optimal duration of omalizumab administration to enhance long-term desensitization while minimizing adverse effects and healthcare costs. Additionally, real-world studies and large-scale RCTs should evaluate the feasibility of integrating omalizumab into routine clinical practice, assessing patient adherence, long-term safety, and sustained tolerance after discontinuation. In addition, leveraging retrospective validation methods using existing datasets can provide critical supplementary evidence to support the broader applicability of omalizumab in food allergy treatment. Future research should prioritize integrating retrospective analyses with ongoing prospective trials to refine treatment strategies and ensure robust clinical applicability across diverse patient populations.

The lack of standardization across studies significantly limits the ability to determine a universally recommended duration for omalizumab administration when combined with OIT. While most studies report a favorable safety profile with omalizumab use for as early as 8 to 12 weeks, the heterogeneity in treatment protocols—ranging from 8 weeks to 28 months—prevents direct comparability. Therefore, future studies should adopt a standardized protocol for omalizumab dosage and duration in combination with OIT to generate more reliable and generalizable data.

Another major gap in the current research is the uncertainty surrounding the long-term sustainability of tolerance following omalizumab discontinuation. Among the studies analyzed, only one assessed long-term sustained unresponsiveness. However, the limited data from this study do not allow for a conclusive assessment of the durability of desensitization, particularly in resource-limited settings where prolonged biologic therapy may not be feasible. To address this, future research should incorporate extended follow-up periods to evaluate the persistence of tolerance and the need for ongoing maintenance therapy.

Furthermore, cost remains a significant barrier to the widespread adoption of omalizumab-facilitated OIT. The financial burden associated with prolonged omalizumab therapy raises concerns regarding its accessibility, particularly in lower-income healthcare settings. Future studies should explore cost-effectiveness by assessing the impact of dose optimization strategies and investigating alternative biologics such as dupilumab, which may offer a more sustainable approach to immune modulation in food allergy treatment.

From a mechanistic perspective, further research is needed to identify specific biomarkers that can predict patient response to omalizumab and OIT. Immunological studies focusing on T-cell regulation, mast cell desensitization, and allergen-specific IgG4 dynamics could provide critical insights into the underlying mechanisms of tolerance induction. Identifying reliable predictive markers would facilitate patient selection, reduce unnecessary exposure to biologic therapy, and improve overall treatment outcomes.

In conclusion, while omalizumab significantly enhances the safety and efficacy of OIT, its widespread clinical implementation is currently hindered by uncertainties related to optimal treatment duration, long-term efficacy, cost-effectiveness, and standardization of treatment protocols. Addressing these challenges through well-designed, large-scale, placebo-controlled trials and mechanistic studies will be crucial in determining the role of omalizumab-facilitated OIT in the management of IgE-mediated food allergies. If these barriers are overcome, this approach has the potential to transform food allergy treatment, improving patient outcomes and quality of life for affected individuals.

## Figures and Tables

**Figure 1 pharmaceuticals-18-00437-f001:**
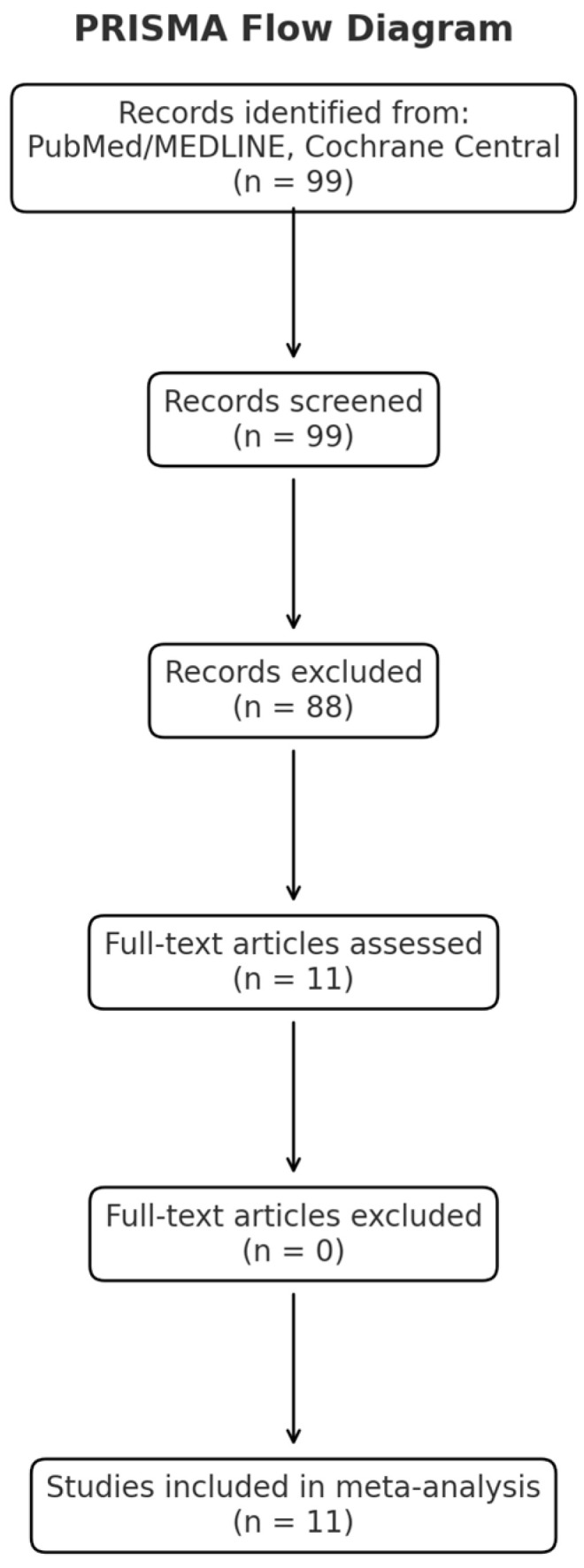
PRISMA flow diagram.

**Table 1 pharmaceuticals-18-00437-t001:** Studies conducted on combined treatment for single-food allergy with oral immunotherapy and omalizumab.

Authors, Year	Study Type N, Age Range	Protocol Description	Success Rate	Immunological Findings	Side Effects
Peanut					
Schneider L.C. et al. 2013 [29]	Monocentric study. N 13 range 8–16 years with a history of IgE-mediated peanut allergy.	Patients were treated with omalizumab for 12 weeks. On week 12, they underwent a rush oral desensitization in 6 h (from 30 to 500 mg; cumulative dose 992 mg peanut flour). For the next 8 weeks (weeks 12–20), subjects returned daily for the slower up-dosing escalation phase of 6 days (from 500 mg to 750 mg) and weekly for each increase to a dose of 4000 mg peanut flour. At week 20, after subjects reached the 4000 mg daily dose, omalizumab was discontinued, but daily oral peanut dosing continued. Twelve weeks after discontinuing the omalizumab (approximately week 32 of the study), a second DBPCFC was conducted from 500 mg to 3500 mg (cumulative dose 8000 mg peanut flour, equivalent to 20 peanuts). If passed, 8000 mg of peanut flour in an open challenge was given 16 h later and 10–20 daily peanuts continued daily.	- All 13 subjects (100%) reached the 500 mg peanut flour dose on the first day (cumulative dose, 992 mg). - Twelve of the thirteen children (92%) reached the 4000 mg dose, requiring a median time of 8 weeks to reach this dose. - In 12 subjects, omalizumab treatment was discontinued after the 4000 mg dose of peanut was achieved and all 12 subjects continued daily peanut dosing (≥4000 mg peanut flour/day) for the rest of the study. - At week 32, 12 subjects underwent a DBPCFC (cumulative dose 8000 mg peanut flour): 11 subjects (85%) tolerated this challenge, and the last subject later passed an open challenge of 8000 mg peanut flour. Twelve of thirteen subjects (92%) tolerated an 8000 mg dose of peanut flour.		- Six of the thirteen subjects (46%) had no or a single mild allergic reaction. - Three subjects (23%) had no allergic reaction during the study. - Six of the thirteen patients (39%) experienced a moderate or severe adverse event. All subjects tolerated omalizumab without adverse reactions, except for occasional injection site pain and swelling.
Brandstrom J. et al. 2019 [27]	One-armed phase 2 study of omalizumab-facilitated peanut OIT. N 23, 12–19 years old, all subjects were peanut-sensitized.	Peanut oral immunotherapy (pOIT) under omalizumab protection started with 280 mg of peanut protein and increased every 2 weeks to 2800 mg (maintenance dose). Eight weeks into the maintenance phase, peanut CD-sensitivity was measured every 8 weeks. If it was suppressed/decreased with no symptoms, omalizumab was reduced with 50%; otherwise, the same dose was given for 8 more weeks; in cases of systemic reactions, omalizumab was increased or the pOIT dose was lowered. OMB was stopped after 8 weeks on 75 mg dose if CD-sensitivity was suppressed with no symptoms, or after 16 weeks if CD-sensitivity was not completely suppressed but stable. POIT was then continued for 12 weeks. Patients passing the single-dose peanut challenge with their current OIT dose were treatment successes (TSs). Those not able to discontinue OMB after 4 years were treatment failures (TFs).	- The primary end-point, pOIT with a daily dose of 2800 mg of peanut protein for 12 weeks after discontinuing omalizumab followed by a negative open challenge with 2800 mg, was met by 9 subjects (39%). - All 23 subjects reached the 2800 mg maintenance dose in a median time of 10 weeks. with significant differences between the treatment success (TS), dropout, and treatment failure (TF) groups; 8, 11.5, and 14 weeks, respectively (*p* = 0.012). Median pOIT duration in TSs was 83 weeks (min–max 48–156) and in TFs was 139 weeks (min–max 115–166).	- From baseline to final visit, no significant differences were seen in IgE: total IgE and IgE-ab to peanut increased slightly, while IgE-ab to control allergens decreased marginally. - SPT for peanut weal diameter decreased significantly in TSs from baseline.	- Mild oropharyngeal and mild abdominal symptoms were reported by 82% and 64%, respectively, in TSs and by 100% and 83%, respectively, in TFs. - A total of 43 systemic reactions: 20 mild, 22 moderate, and 1 severe.
MacGinnitte et al. 2017 [28]	Randomized placebo-controlled study. N 37, median age 10 years old.	Thirty-seven subjects were randomized to omalizumab (*n* = 29) or placebo (*n* = 8). After 12 weeks of treatment with omalizumab, children underwent a rapid one-day desensitization of up to 250 mg of peanut protein, followed by weekly increases up to 2000 mg. Omalizumab was then discontinued and subjects continued on 2000 mg of peanut protein. They underwent an open challenge to 4000 mg peanut protein twelve weeks after stopping the study drug. If tolerated, subjects continued on 4000 mg of peanut protein daily.	The median peanut dose tolerated on the initial desensitization day was 250 mg for omalizumab versus 22.5 mg for placebo-treated subjects. Subsequently, 23 of 29 (79%) subjects randomized to omalizumab tolerated 2000 mg peanut protein 6 weeks after stopping omalizumab, versus 1 of 8 (12%) receiving a placebo (*p* < 0.01). Twenty-three subjects on omalizumab versus 1 on placebo passed the 4000 mg food challenge. Omalizumab allows subjects with peanut allergy to be rapidly desensitized over as little as 8 weeks of peanut OIT. In the majority of subjects, this desensitization was sustained after omalizumab was discontinued.	Subjects treated with omalizumab showed decreased wheal size on skin testing but increased peanut-specific IgE values at week 31 compared to baseline. The single placebo-treated subject who tolerated the 4000 mg food challenge showed increases in both peanut-specific IgE and SPT.	Overall reaction rates were not significantly lower in omalizumab versus placebo-treated subjects (OR = 0.57 *p* = 0.15), although omalizumab-treated subjects were exposed to much higher doses of peanuts.
Milk					
Badina et al. 2022 [33]	N 4 patients (8–24 years old) who had cow milk allergy, asthma, suspended previous OIT for milk, and reacted to an amount of less than 173 mg of milk.	The patients, after a course of 8 weeks of OMB, underwent an OFC to reevaluate the threshold. Then, a new OIT was performed using the same protocol of the previous attempts, maintaining therapy with OMB for 12 months (at the suspension of OMB, the maintaining dose was reduced by 30%).	During OIT, the four patients experienced no reactions or extremely mild ones (oral itching or transient mild abdominal pain).	All increased their threshold of CM in OML compared with the baseline and maintained it long after biologic therapy had discontinued. Specific milk proteins’ IgG4 levels significantly increased in all.	- Two months after the suspension of Omalizumab, one patient restarted the therapy for deterioration in control of asthma. - One patient decided to reduce the intake of milk during the maintenance phase due to anxiety (no adverse reaction).
Nadeau et al. 2011 [30]	N 11, 7–17 years old, all, all subjects had milk allergy	After 9 weeks of omalizumab treatment, on the first day a rush oral desensitization was performed (starting with 0.1 mg of milk powder up to 1000 mg, with increasing doses every 30 min). Then, desensitization with daily doses of milk was continued for the next 7 to 11 weeks (all dose increases were administered in the clinical research unit and, if tolerated, were given at home). Omalizumab was discontinued at 16 weeks, while daily oral milk consumption was continued; then, a double-blind, placebo-controlled food challenge (DBPCFC) with 5 doses of milk or placebo was conducted at week 24 of the study.	- One patient discontinued the study due to abdominal migraine. - In total, 9/10 patients achieved tolerance to 1000 mg of milk during the first day of the rush oral desensitization. - After weekly increases in the doses of milk, 9/10 patients achieved tolerance to 2000 mg of milk.		All patients experienced some adverse effects; most were mild and did not need treatment. During the rush phase, - One patient discontinued the study due to abdominal migraine. - After the administration of the 1000 mg dose, 1 subject received epinephrine for nasal obstruction and generalized urticaria refractory to diphenhydramine and cetirizine. - One patient had tongue swelling which responded to antihistamines. At home, during the maintenance phase, - Two patients were given epinephrine: one for urticaria of the upper left leg and tongue swelling; one for urticaria of the right upper arm.
Takahashi et al. 2017 [31]	Prospective randomized controlled trial. N 16, age 6–14 years, with high IgE levels to CM.	Patients were randomized 1:1 to an OMB-OIT group or untreated group. OIT was done 8 weeks from the start of OMB treatment. The initial dose of OIT was set at a sub-threshold dose that was usually one-tenth of the threshold dose. After the initial dose, the next and subsequent doses were increased approximately 1.5-fold until the threshold dose was reached. After reaching the threshold dose, the next and subsequent doses were increased approximately 1.2-fold. The target dose of CM was 200 mL of MH-CM. If there were no further increases in dose because of repeated adverse events, then escalation of OIT using fresh CM was started after the highest tolerated dose was continued for 3 consecutive days without an allergic reaction. If the target dose was achieved, dose escalation was terminated and that dose was chosen as the maintenance phase. OMB was discontinued 24 weeks from the start of its administration.	The primary outcome was the induction of desensitization at 8 weeks after OMB was discontinued in the OMB-OIT-treated group and at 32 weeks after study entry. Secondary outcomes included changes in the OFC SCD, incidence of OIT-related adverse reactions, changes in the SPT, total IgE levels, CM-related sIgE levels, CM-related sIgGs, and CM-related sIgA. None of the 6 children in the untreated group developed desensitization to CM, while all of the 10 children in the OIT-OMB-treated group achieved desensitization.	There was no significant difference in levels of total IgE, CM sIgE, casein sIgE, β-lactoglobulin sIgE, or α-lactoalbumin sIgE between the two groups at week 32.	Severe adverse events were not observed, even during the escalation phase. No epinephrine injections were given in either phase. Anaphylactic shock and death induced by CM ingestion were not observed in either group during the study.
Wood et al. 2016 [32]	Double-blind, placebo-controlled trial with subjects randomized to omalizumab or placebo. N 57 (7–32 years), randomized milk-specific IgE, skin tests or OFCs.	Open-label MOIT was initiated after 4 months of omalizumab/placebo with escalation to maintenance over 22–40 weeks, followed by daily maintenance dosing through month 28. At month 28, omalizumab was discontinued and subjects passing an OFC with 10 g continued OIT for 8 weeks, after which OIT was discontinued, with a re-challenge at month 32 to assess sustained unresponsiveness.	At month 28, 24 omalizumab-treated subjects and 20 placebo-treated subjects passed the desensitization OFC. At month 32, sustained unresponsiveness was demonstrated in 48.1% in the omalizumab group and 35.7% in the placebo group (*p* = 0.42). Adverse reactions were markedly reduced during OIT escalation in omalizumab subjects for percent doses/subject-provoking symptoms, dose-related reactions requiring treatment, and doses required to achieve maintenance.	Significant increases from baseline of casein and beta-lactoglobulin IgG4 bevels were detected within both treatment groups from month 16 onward, with no differences seen between the two groups. In the omalizumab group, milk- and casein-specific IgE were significantly increased at month 4 and reduced at month 32, while in the placebo group, all milk and casein IgE levels were significantly reduced after month 4.	Reactions requiring epinephrine tended to be more common in the placebo group. When considering all subjects, 20 of 40,641 total doses led to reactions requiring epinephrine in 11 individuals, with 2 doses in 2 omalizumab-treated subjects and 18 doses in 9 placebo-treated subjects.

**Table 2 pharmaceuticals-18-00437-t002:** Studies conducted on combined treatment for multi-food allergies with oral immunotherapy and omalizumab.

Authors, Year	Study Type N, Age Range	Protocol Description	Success Rate	Immunological Findings	Side Effects
Multi-food allergies, trial in progress					
Andorf S. et al. 2019 [34]	Phase 2 randomized controlled multisite study. N 70, aged 5–22 years, with multiple food allergies.	In the open-label phase, all participants received omalizumab (weeks 1–16) and multi-OIT (2–5 allergens; weeks 8–30). At weeks 28–29, all participants on a maintenance dose (60) of each allergen were randomized 1:1:1 to 1 g, 300 mg, or 0 mg arms (blinded, weeks 30–36) and then tested by food challenge at week 36. Success was defined as passing a 2 g food challenge to at least 2 foods in week 36. The primary outcome was comparing the primary endpoint between the combined active treatment arms versus the 0 mg arm. Secondary endpoints included the comparison between the percent of participants who passed a food challenge at week 36 to at least 2 g and at least 4 g of each of food allergens (of at least 2, 3, 4, or all 5 foods).	In total, 17/17 of 19 randomized to the 1 g arm, 17/20 of 21 randomized to the 300 mg arm, and 11/16 of 20 randomized to the 0 g arm (discontinuation arm) tolerated a dose of 2 g or higher in the week 36 food challenges. The PP participants showed a difference between the combined treatment (1 g and 300 mg arms) and the 0 mg arm in achieving the primary endpoint (*n* = 34/37, 92% vs. *n* = 11/16, 69%, *p* = 0.045), but demonstrated no significant differences between the active treatment arms (1 g vs. 300 mg) (*n* = 17/17, 100% vs. *n* = 17/20, 85%, OR: 0; 95% CI: 0–2.8; *p* = 0.23). The active treatment arms were not more likely to successfully pass an OFC to 4 g protein of at least 2 food allergens at week 36 compared to the 0 mg discontinuation arm (70% vs. 45%, *p* = 0.09). The number of food allergens (2, 3, 4, or 5) in the active arms was significantly higher than in discontinuation arm (*p* = 0.02).	In all three study arms, no significant difference in peanut-specific IgE at baseline was detected between participants that met or failed the primary endpoint containing peanut in their multi-OIT; a significant increase in peanut-specific IgG4/IgE from baseline to week 36 can be seen. Independent of the outcome of the food challenge at week 36, the SPT wheal did not show a significant difference for any food, stratified by the three randomization groups, between week 30 and week 36.	A higher number of doses of injectable epinephrine occurred in the 1 g (4 doses of injectable epinephrine) vs. discontinuation (1 dose of injectable epinephrine) vs. 300 mg (no doses of injectable epinephrine) arms. No cases of life-threatening anaphylaxis or eosinophilic esophagitis occurred during the study.
Wood R. et al. 2022 [35]	Phase 3, multicenter, randomized, double-blind, placebo-controlled study. N 177, aged 1–17 years, with peanut allergy and at least 2 other food allergies (including milk, egg, wheat, cashew, hazelnut, or walnut).	The OUtMATCH study consisted of 3 stages. In stage 1, 177/462 patients, who reacted to <100 mg of peanut protein (cumulative dose 144 mg) and <300 mg of protein (cumulative dose 444 mg) for each of the other 2 foods during DBPCFC, were randomized 2:1 to receive 16 to 20 weeks of treatment with omalizumab or a placebo and, after 16 weeks of treatment, repeated a DBPCFC to each of 3 foods for a cumulative dose of 6044 mg of protein of each food. The first 60 participants who completed stage 1 were assigned to a 24-week open label extension of omalizumab, followed by a DBPCFC to each of their 3 foods and a placebo (cumulative dose of 8044 mg of protein of each food) and then passed to stage 3. All the other participants passed to stage 2 (52 weeks of active or placebo OIT), received 8 weeks of treatment with open label omalizumab, and then were randomized to double-blind treatment with either (1) omalizumab-facilitated OIT for 8 weeks followed by placebo-multiallergen OIT for 44 weeks or (2) omalizumab-placebo OIT for 8 weeks followed by omalizumab-placebo OIT for 44 weeks. At the end of 52 weeks, a DBPCFC to each of their 3 foods and a placebo were performed (cumulative dose of 8044 mg of protein of each food); if passed, they moved to stage 3. In stage 3, each participant received a long-term follow-up with dietary consumption of a food, or with avoidance of a food, or rescue OIT for a food, depending on the participant’s preferences, for 12–36 months. Primary end point was tolerance of >600 mg of peanut protein and >1000 mg of cashew, milk, or egg protein (DBPCFC at the end of stage 1), then >2000 mg of protein of all 3 foods (DBPCFC at the end of stage 2) and the dietary consumption of foods at the end of stage 3 compared to earlier stages.	In total, 79/118 (64%) participants receiving omalizumab reached the primary endpoint, as compared with 4/59 (7%) of the placebo arm (*p* < 0.001). Overall, 66% of patients receiving omalizumab were able to consume milk, 67% egg, 41% cashew, 64% walnut, 65% hazelnut, and 75% wheat, compared to 10%, 0%, 3%, 13%, 14%, and 13% for the placebo arm, out of 62, 71, 99, 78, 24, and 20 individuals, respectively.		Safety end points did not differ between the groups, aside from more injection-site reactions in the omalizumab group.
Begin P. et al. 2014 [36]	Open-label, phase 1, single-site OIT protocol. N 25 (median age 7 years) with multiple food allergies.	Participants received OIT for up to 5 allergens simultaneously with omalizumab (rush mOIT). On the initial escalation day, dosing began at 5 mg total food allergen protein divided equally between each of the offending foods and doses were slowly increased until the participant reached a final dose of 1250 mg protein. The participants returned to the CTFU every two weeks for a dose escalation built with an individualized schedule to reach 4000 mg dose per food allergen protein. Omalizumab was administered for 8 weeks prior to and 8 weeks following the initiation of the rush mOIT schedule (16 weeks in total).	Overall, 19/22 (76%) participants tolerated all 6 steps of the initial escalation day (up to 1250 mg of combined food proteins). The remaining 6 were started on their highest tolerated dose. In total, 3/25 withdrew. All 22/22 were able to reach a maintenance dose (4000 mg per allergen), with a median time to reach of 18 weeks (7–36 weeks) with all participants.	After 52 weeks of therapy, peanut-specific IgE (PN-IgE) did not change significantly. However, peanut- specific IgG4 (PN-IgG4) levels showed median increases of 8.23 mgA/L (*p* < 0.0001), while peanut SPT decreased by a median of 8 mm (*p* < 0.0001), at baseline and after a year of therapy for participants with proven peanut allergy.	For the initial dose escalation day, dose escalations, and home dosing, most (94%) allergic reactions were mild; there were no serious adverse events, although 13 participants (52%) experienced some symptoms on their initial dose escalation day. One severe reaction occurred at home shortly after reaching the maintenance phase (16,000 mg) in a participant desensitized to peanut, almond, milk, and egg.
Langlois A. et al. 2020 [26]	Multicenter, phase 2b, double-blind, randomized controlled clinical trial. N 90, aged 6 to 25, with 3 or more food allergies.	Ninety participants were randomized 2:2:1 to receive 20 weeks of omalizumab at monthly dosages of 16 mg/kg (*n* = 36), 8 mg/kg (*n* = 36), or a placebo (*n* = 18). The study drug was given at full dosage for a total of 12 weeks with a progressive taper during the last 8 weeks. Multi-food OIT was started after a pre-treatment period of 8 weeks, with biweekly up-dosing according to a symptom-driven schedule until the target dose of 1500 mg of food protein was reached (500 mg per food). The primary endpoint was the time for reaching the target maintenance dose, compared between the 3 arms. Secondary endpoints were the change in reactivity threshold to foods after pre-treatment with study drug; average up-dosing speed; and adverse reaction rate.	The sample size was be primarily driven by the 16 vs. 8 mg/kg comparison. Assuming a median time-to-maintenance of 2 weeks in the 16 mg/kg arm, based on the clinical cohort, a sample of 72 participants (36 in each arm) would confer a power of 0.80 to detect a HR = 2.2 of time-to- maintenance with an alpha of 0.017 (considering 3-way testing between the study arms), assuming administrative censoring at 52 weeks. A HR of 2.15 would mean 2.3 additional OIT weeks, which was considered the minimal clinically relevant difference (i.e., one up-dosing visit). Assuming a median time-to-maintenance of 6 weeks in the 8 mg/kg arm, a sample of 54 participants (18 in placebo arm) would confer a power of 0.80 to detect a HR = 2.54, given a time-to-maintenance with an alpha of 0.017, assuming administrative censoring at 52 weeks. A HR of 2.54 would mean 9.2 additional OIT weeks, which was considered the minimal clinically relevant difference to consider adding adjunct drug therapy.		

## Data Availability

Data are contained within the article and Appendix A.

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
