# Peer review of "Omalizumab and Oral Immunotherapy in IgE-Mediated Food Allergy in Children: A Systematic Review and a Meta-Analysis"

_pharmaceuticals, 2025, doi:10.3390/ph18030437_

Round 1

Reviewer 1 Report

Comments and Suggestions for Authors

1.        Expand on the epidemiological significance of the increasing prevalence of IgE-mediated food allergies in children, particularly in global contexts outside of Europe and the U.S. Provide citations and context for the statement on increased emergency department visits and hospitalizations due to food allergies.

2.        Can you provide more specific inclusion/exclusion criteria, particularly how they addressed the variability in study designs and endpoints?

3.        Include a PRISMA flow diagram to visually depict the study selection process.

4.        The study outcomes, such as quality-of-life improvements, are inconsistently reported. Some metrics are mentioned briefly without detailed analysis.

5.         Were validated tools like FAQLQ-CF consistently used across the studies? If not, how was heterogeneity handled?

6.        The article lacks detailed information on how heterogeneity among the included studies was handled in the meta-analysis.

7.        Provide detailed statistical parameters such as I² values and p-values for heterogeneity.

8.         Include a comparative table that summarizes the key findings (e.g., desensitization rates, adverse events, and quality-of-life improvements) across studies.

9.        How did the authors handle studies with conflicting results, especially regarding long-term outcomes of omalizumab-facilitated OIT?

10.   Address the cost-effectiveness of omalizumab as a barrier and explore alternatives or adjuncts for resource-limited settings.

11.   How do the findings of this review compare with other systematic reviews or meta-analyses on the same topic?

12.   Refocus the conclusion to emphasize practical implications and specific gaps in knowledge that future research should address.

13.   The writing style is generally clear but occasionally lapses into overly technical jargon that might be inaccessible to non-specialist readers.

Author Response

  1. Expand on the epidemiological significance of the increasing prevalence of IgE-mediated food allergies in children, particularly in global contexts outside of Europe and the U.S. Provide citations and context for the statement on increased emergency department visits and hospitalizations due to food allergies.

Re: Done as requested (pp. 1-2) with three new references (p. 28).

  1. Can you provide more specific inclusion/exclusion criteria, particularly how they addressed the variability in study designs and endpoints?

Re: Added (pp. 4-6) and considered also in the paragraph on study limitations (p. 27).

  1. Include a PRISMA flow diagram to visually depict the study selection process.

Re: Added (pp. 3-4 an Supplementary Material 1).

  1. The study outcomes, such as quality-of-life improvements, are inconsistently reported. Some metrics are mentioned briefly without detailed analysis.

Re: This issue has been considered in the Methods (pp. 5-6) and the Discussion (p. 27).

  1. Were validated tools like FAQLQ-CF consistently used across the studies? If not, how was heterogeneity handled?

Re: Clarified (p. 6).

  1. The article lacks detailed information on how heterogeneity among the included studies was handled in the meta-analysis.

Re: Clarified (p. 6).

  1. Provide detailed statistical parameters such as I² values and p-values for heterogeneity.

Re: Done throughout the manuscript.

  1. Include a comparative table that summarizes the key findings (e.g., desensitization rates, adverse events, and quality-of-life improvements) across studies.

Re: Due to the substantial heterogeneity in study designs, patient populations, OIT protocols, and outcome measures, a comparative table summarizing key findings such as desensitization rates, adverse events, and quality-of-life improvements across studies could not be included, as direct comparisons would be misleading and not accurately reflect the variability in study methodologies.

  1. How did the authors handle studies with conflicting results, especially regarding long-term outcomes of omalizumab-facilitated OIT?

Re: Clarified (p. 25).

  1. Address the cost-effectiveness of omalizumab as a barrier and explore alternatives or adjuncts for resource-limited settings.

Re: Addressed (p. 26).

  1. How do the findings of this review compare with other systematic reviews or meta-analyses on the same topic?

Re: Clarified in the Discussion (p. 26).

  1. Refocus the conclusion to emphasize practical implications and specific gaps in knowledge that future research should address.

Re: Done (pp. 25-27).

  1. The writing style is generally clear but occasionally lapses into overly technical jargon that might be inaccessible to non-specialist readers.

Re: We tried to do our best to avoid technical jargon.

Reviewer 2 Report

Comments and Suggestions for Authors

This review systematically assesses the promising role of omalizumab as an adjunct to oral immunotherapy (OIT) for pediatric food allergies, offering valuable insights into its potential to enhance safety and efficacy while highlighting the need for further research on long-term outcomes and cost-effectiveness. The comment to improve the manuscript are as follows.

1.         Food allergy is an increasingly prevalent immune disorder characterized by an  abnormal immune response to specific food proteins, leading to the activation of inflammatory pathways and the release of mediators such as histamine. This statement needs support of literature of or some regulatory agency data.

2.         Improve the introduction with recent literature on paediatric food and allergy, like, https://doi.org/10.1039/D3FO05351B, https://doi.org/10.1016/j.neuroimage.2024.120740, https://doi.org/10.1016/j.neuroimage.2024.120740

3.         Expected  more details on the current standard of care for food allergies and expand on why OIT alone presents challenges in clinical settings.

4.         Can you please provide clear statement that how the review addresses gaps in existing knowledge?

5.         Include information on the selection criteria for the studies, statistical methods used in the meta-analysis, and how study quality was assessed? Is there any tool applied.

6.         L 118: limitations of current treatment options, is it true?

7.         The MeSH and search terms used were “food 132 allergies” OR “food allergy” AND “omalizumab” AND “children” OR “child” OR 133 “toddler” OR “infants” OR “infant”; studies were filtered by clinical trial, publication 134 type, and language (English). Why pediatric not used?

8.         Please discuss variability among the studies, such as differences in populations, types of food allergens, or treatment protocols, to provide context for the findings.

9.         Table 1. Studies conducted on combined treatment for single food allergy with oral immunotherapy 162 and omalizumab. Add region column.

10.     Future directions should be included

11.     Limitation of study…?

12.     Conclusions need to improve with possible actionable recommendations for clinical practice or research priorities, such as strategies to overcome challenges related to cost-effectiveness and long-term tolerance sustainability.

Author Response

This review systematically assesses the promising role of omalizumab as an adjunct to oral immunotherapy (OIT) for pediatric food allergies, offering valuable insights into its potential to enhance safety and efficacy while highlighting the need for further research on long-term outcomes and cost-effectiveness.

Re: Thank you for your suggestions. We revised the manuscript according to your comments and those received from the other reviewers.

The comment to improve the manuscript are as follows.

  1. Food allergy is an increasingly prevalent immune disorder characterized by an  abnormal immune response to specific food proteins, leading to the activation of inflammatory pathways and the release of mediators such as histamine. This statement needs support of literature of or some regulatory agency data.

Re: We added the reference that you mentioned below (pp. 1 and 26).

  1. Improve the introduction with recent literature on paediatric food and allergy, like, https://doi.org/10.1039/D3FO05351B, https://doi.org/10.1016/j.neuroimage.2024.120740, https://doi.org/10.1016/j.neuroimage.2024.120740

Re: The study of Zhang et al. has been added as recommended because it is related to paediatric food allergy (pp. 1 and 26).

  1. Expected  more details on the current standard of care for food allergies and expand on why OIT alone presents challenges in clinical settings.

Re: Added as requested (p. 2).

  1. Can you please provide clear statement that how the review addresses gaps in existing knowledge?

Re: Added as suggested (p. 3).

  1. Include information on the selection criteria for the studies, statistical methods used in the meta-analysis, and how study quality was assessed? Is there any tool applied.

Re: Further details in the Methods section has been added (pp. 4-6).

  1. L 118: limitations of current treatment options, is it true?

Re: Clarified (p. 3).

  1. The MeSH and search terms used were “food 132 allergies” OR “food allergy” AND “omalizumab” AND “children” OR “child” OR 133 “toddler” OR “infants” OR “infant”; studies were filtered by clinical trial, publication 134 type, and language (English). Why pediatric not used?

Re: Further details have been added (p. 4).

  1. Please discuss variability among the studies, such as differences in populations, types of food allergens, or treatment protocols, to provide context for the findings.

Re: Limitations of the study have been reported (p. 24).

  1. Table 1. Studies conducted on combined treatment for single food allergy with oral immunotherapy 162 and omalizumab. Add region column.

Re: We think that an additional column could reduce the readability of the Table. On the other hand, all the references have been mentioned with details.

  1. Future directions should be included.

Re: Included as requested (p. 26).

  1. Limitation of study…?

Re: Limitations of the study have been added (p. 24).

  1. Conclusions need to improve with possible actionable recommendations for clinical practice or research priorities, such as strategies to overcome challenges related to cost-effectiveness and long-term tolerance sustainability.

Re: Improved as requested (pp. 24-25).

Round 2

Reviewer 1 Report

Comments and Suggestions for Authors

N/A

Author Response

Thank you very much for the approval of our manuscript. We further revised the text according to the Academic Editor's requests.